# A SURGERY OF THE NEURAL ARCHITECTURE EVALUATORS

## ABSTRACT

Neural architecture search (NAS) has recently received extensive attention due to its effectiveness in automatically designing effective neural architectures. A major challenge in NAS is to conduct a fast and accurate evaluation (i.e., performance estimation) of neural architectures. Commonly used fast architecture evaluators include parameter-sharing ones and predictor-based ones. Despite their high evaluation efficiency, the evaluation correlation (especially of the well-performing architectures) is still questionable. In this paper, we conduct an extensive assessment of both the parameter-sharing and predictor-based evaluators on the NAS-Bench-201 search space, and break up how and why different configurations and strategies influence the fitness of the evaluators. Specifically, we develop a set of NAS-oriented criteria to understand the behavior of fast architecture evaluators in different training stages. And based on the findings of our experiments, we give pieces of knowledge and suggestions to guide NAS application and motivate further research.

## 1 INTRODUCTION

Studies have shown that the automatically discovered architectures by NAS can outperform the hand-crafted architectures for various applications, such as classification (Nayman et al., 2019; Zoph & Le, 2017), detection (Ghiasi et al., 2019; Chen et al., 2019b), video understanding (Ryoo et al., 2019), text modeling (Zoph & Le, 2017), etc. Early NAS algorithms (Zoph & Le, 2017) suffer from the extremely heavy computational burden, since the evaluation of neural architectures is slow. Thus, how to estimate the performance of a neural architecture in a fast and accurate way is vital for addressing the computational challenge of NAS.

A neural architecture evaluator outputs the evaluated score of an architecture that indicates its quality. The straightforward solution is to train an architecture from scratch to convergence and then test it on the validation dataset, which is extremely time-consuming. Instead of exactly evaluating architectures on the target task, researchers usually construct a *proxy model* with fewer layers or fewer channels (Pham et al., 2018; Real et al., 2019; Wu et al., 2019), and train this model to solve a *proxy task* of smaller scales (Cai et al., 2018a; Elsken et al., 2018; Klein et al., 2017; Wu et al., 2019), e.g., smaller dataset or subsets of dataset, training or finetuning for fewer epochs.

Traditional evaluators conduct separate training phases to acquire the weights that are suitable for each architecture. In contrast, *one-shot evaluation* amortized the training cost of different architectures through parameter sharing or a global hypernetwork, thus significantly reduce the evaluation cost. Pham et al. (2018) constructs an over-parametrized super network (supernet) such that all architectures in the search space are sub-architectures of the supernet. Throughout the search process, the shared parameters in the supernet are updated on the training dataset split, and each architecture is evaluated by directly using the corresponding subset of the weights. Afterwards, the parameter sharing technique is widely used for architecture search in different search spaces (Wu et al., 2019; Cai et al., 2020), or incorporated with different search strategies (Liu et al., 2018b; Nayman et al., 2019; Xie et al., 2018; Yang et al., 2019; Cai et al., 2020). Hypernetwork (Brock et al., 2018; Zhang et al., 2018) based evaluation is another type of one-shot evaluation strategy, in which a hypernetwork is trained to generate proper weights for each architecture. Since hypernetwork solutions are not generic currently, this paper concentrates on the evaluation of parameter sharing evaluators.

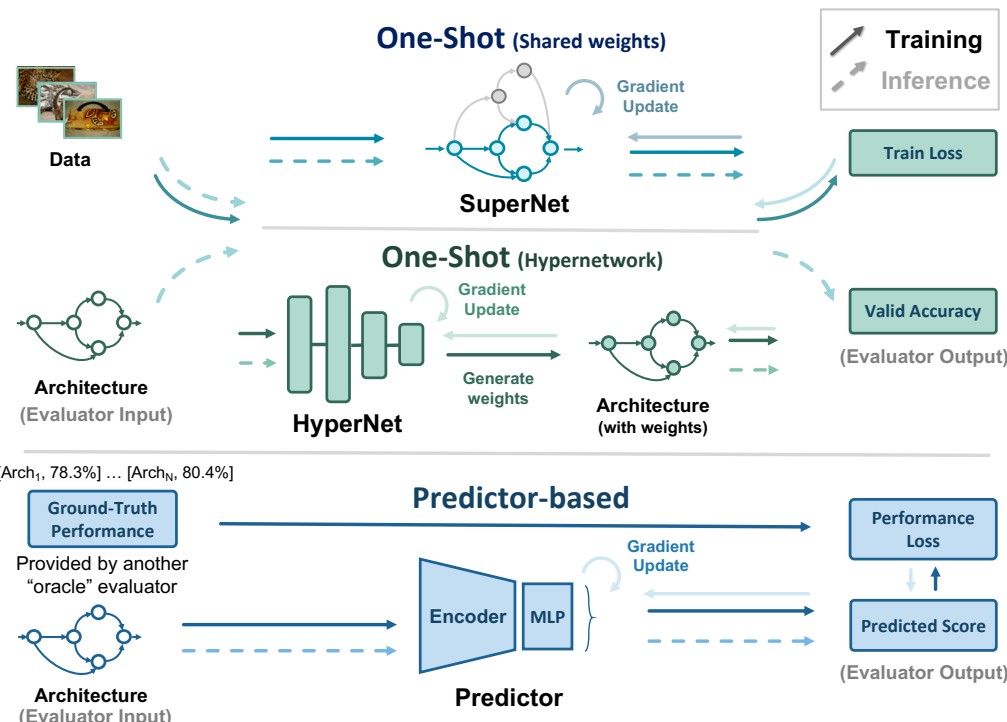

Figure 1: An overview of fast neural architecture evaluators (i.e., performance estimators).

Whether or not one-shot strategies can provide highly-correlated architecture evaluation results is essential for the efficacy of the NAS process. Many recent studies have been focusing on assessing the evaluation correlation of one-shot architecture evaluators (Bender et al., 2018; Sciuto et al., 2019; Zela et al., 2020).

Besides one-shot evaluation strategies, *predictor-based evaluation* strategies (Luo et al., 2018; Liu et al., 2018a; Deng et al., 2017; Sun et al., 2019; Wang et al., 2018; Xu et al., 2019; Ning et al., 2020) use a performance predictor that takes the architecture description as inputs and outputs a predicted performance score. The performance predictor should be trained using "ground-truth" architecture performances. This paper utilizes the same set of criteria to evaluate and compare different performance predictors.

The fast neural architecture evaluators (i.e., performance estimators) are summarized in Fig. 1, including parameter sharing, hypernetworks, and predictor-based ones. And this paper aims at revealing the status of current architecture evaluation strategies systematically. Specifically, we develop a set of NAS-oriented criteria to understand the behavior of fast architecture evaluators in different training stages. And based on the findings of our experiments, we give pieces of knowledge and suggestions to guide NAS application and motivate further research.

The knowledge revealed by this paper includes: 1) Layer proxy brings a larger evaluation gap than using channel proxy, thus channel proxy can be utilized to reduce the computational cost, while proxy-less search w.r.t the layer number is worth studying. 2) The convergence rate of different criteria varies during the one-shot supernet training, which shows that the good architectures are distinguished from bad architectures in the early stage. 3) As training goes on, the one-shot performances of isomorphic sub-architectures become closer. 4) De-isomorphic sampling or post de-isomorphism handling can help avoid the over-estimation of simple architectures. 5) Parameter sharing evaluator tends to over-estimate smaller architectures, and is better at comparing smaller models than larger models. 6) One should use ranking losses rather than regression losses to train predictors, since they are more stable. 7) Different predictors under- or over-estimate different architectures, and currently, the best predictor might still have trouble in comparing large architectures. 8) As expected, architecture predictors can distinguish good architectures better after multiple stages of training, as the training data are more and more concentrated on the good architectures.

## 2 RELATED WORK

### 2.1 ONE-SHOT EVALUATORS

One-shot evaluation mainly consists of two types of strategies: 1) parameter sharing (Pham et al., 2018; Wu et al., 2019; Liu et al., 2018b; Nayman et al., 2019; Xie et al., 2018; Yang et al., 2019; Cai et al., 2020), 2) hypernetworks (Brock et al., 2018; Zhang et al., 2018). These two strategies both amortize the training cost of different architectures via the sharing of the network or hypernetwork parameters.

The ranking correlation gaps of existing shared weights evaluators are brought by two factors: 1) proxy model and task: due to the memory constraint, a proxy supernet (supernet) (Liu et al., 2018b; Pham et al., 2018) with fewer channels or layers is usually used; 2) parameter sharing. To alleviate the first factor, there are some studies (Cai et al., 2018b; Chen et al., 2019a) that aim at making one-shot evaluation more memory efficient, thus the one-shot search could be conducted without using a proxy supernet. As for the second factor, there are a few studies that carried out correlation evaluation for one-shot evaluators. Zhang et al. (2018) conducted a correlation comparison between the GHN hypernetwork evaluator, shared weights evaluator, and several small proxy tasks. However, the correlation is evaluated using 100 architectures randomly sampled from a large search space, which is not a convincing and consistent benchmark metric. Luo et al. (2019) did a preliminary investigation into why parameter sharing evaluation fails to provide correlated evaluations, and proposed to increase the sample probabilities of the large models. Their evaluation is also conducted on dozens of architectures sampled from the search space. Zela et al. (2020) compare the evaluation correlation of different search strategies on NAS-Bench-101. Sciuto et al. (2019) conduct parameter sharing NAS in a toy RNN search space with only 32 architectures in total, and discover that the parameter sharing rankings do not correlate with the true rankings of architectures. To improve the evaluation correlation, Chu et al. (2019) proposed a sampling strategy in a layer-wise search space.

In this paper, we analyze the ranking correlation gaps brought by the model proxy (difference in the number of channels and layers) and the parameter sharing technique, as well as the behavior of one-shot evaluators during the training process.

### 2.2 PREDICTOR-BASED EVALUATORS

An architecture performance predictor takes the architecture descriptions as inputs, and outputs the predicted performance scores without training the architectures. Two factors are crucial to the fitness of the predictors: 1) embedding space; 2) training technique. On the one hand, to embed neural architectures into a continuous space and get a meaningful embedding space, there are studies that propose different architecture encoders, e.g., sequence-based (Luo et al., 2018; Liu et al., 2018a; Deng et al., 2017; Sun et al., 2019; Wang et al., 2018), graph-based (Shi et al., 2019; Ning et al., 2020). As for nonparametric predictors, Kandasamy et al. (2018) design a kernel function in the architecture space and exploits gaussian process to get the posterior of the architecture performances. Shi et al. (2019) combined a graph-based encoder and nonparametric gaussian process to construct the performance predictor. On the other hand, from the aspect of training techniques, Luo et al. (2018) employed an encoder-decoder structure and used an auxiliary reconstruction loss term. Xu et al. (2019); Ning et al. (2020) employed learning-to-rank techniques to train the predictors.

Actually, in the overall NAS framework, the predictor-based evaluator plays a different role from the traditional or one-shot evaluators, since the predictor should be trained using "ground-truth" architecture performances. Usually, expensive traditional evaluators that can provide relatively accurate architecture performances are chosen as the "oracle" evaluators to output the "ground-truth" scores (Kandasamy et al., 2018; Liu et al., 2018a; Luo et al., 2018).

## 3 EVALUATION CRITERIA

In this section, we introduce the evaluation criteria used in this paper. We denote the search space size as $M$, the true performances and approximated evaluated scores of architectures $\{a_i\}_{i=1,\cdots,M}$ as $\{y_i\}_{i=1,\cdots,M}$ and $\{s_i\}_{i=1,\cdots,M}$, respectively. And we denote the ranking of the true performance

$y_i$ and the evaluated score $s_i$ as $r_i \in \{1, \cdots, M\}$ and $n_i \in \{1, \cdots, M\}$ ($r_i = 1$ indicates that $a_i$ is the best architecture in the search space). The correlation criteria adopted in our paper are

- Linear correlation: The pearson correlation coefficient $\text{corr}(y, s)/\sqrt{\text{corr}(y, y)\text{corr}(s, s)}$.
- Kendall's Tau ranking correlation: The relative difference of concordant pairs and discordant pairs $\sum_{i<j} \text{sgn}(y_i - y_j)\text{sgn}(s_i - s_j)/\binom{M}{2}$.
- Spearman's ranking correlation: The pearson correlation coefficient between the rank variables $\text{corr}(r, n)/\sqrt{\text{corr}(r, r)\text{corr}(n, n)}$.

Besides these correlation criteria, criteria that emphasize more on the relative order of architectures with good performances are desired. Denoting $A_K = \{a_i | n_i < KM\}$ as the set of architectures whose evaluated scores $s$ is among the top $K$ portion of the search space, we use two criteira

- Precision@K (P@K) $\in (0, 1] = \frac{\#\{i | r_i < KM \land n_i < KM\}}{KM}$: The proportion of true top-K proportion architectures in the top-K architectures according to the scores.
- BestRanking@K (BR@K) $\in (0, 1] = \arg\min_{i \in A_K} r_i$: The best normalized ranking among the top K proportion of architectures according to the scores.

The two criteria are similar to those used in Ning et al. (2020), except that rankings and architecture numbers are all normalized with respect to the search space size $M$.

The above criteria are used to compare the fitness of various architecture evaluators with different configurations and in different stages. Besides that, we'd also like to interpret their evaluation results. To identify which architectures are under- or over-estimated by various evaluators, and analyze the reasons accordingly, we investigate the relationship of the true-predicted ranking differences $\{r_i - n_i\}_{i=1,\cdots,M}$ and the architecture properties such as the FLOPs: $\{\text{FLOPs}(a_i)\}_{i=1,\cdots,M}$.

## 4 PARAMETER-SHARING EVALUATORS

In this section, we assess the behavior of one-shot evaluators and evaluate the influence of several sampling strategies and training techniques.

### 4.1 EXPERIMENTAL SETUP

During the supernet training process, candidate architectures are randomly sampled, and their corresponding weights are updated in every iteration.

We conduct our experiments on CIFAR-10 using a recent NAS benchmarking search space, NAS-Bench-201 (Dong & Yang, 2020). NAS-Bench-201 is a NAS benchmark that provides the performances of all the 15625 architectures in a cell-based search space. Actually, there are architectures with different matrix representations that are isomorphic (i.e., represent the same architecture) in this search space. As reported by their original paper, there are 6466 unique topology structures after de-isomorphism.

The hyper-parameters used to train all the parameter-sharing supernets are summarized in Tab. 1. We train parameter sharing evaluators via momentum SGD with momentum 0.9 and weight decay 0.0005. The batch size is set to 512. The learning rate is set to 0.05 initially and decayed by 0.5 each time the supernet accuracy stops to increase for 30 epochs. During training, the dropout rate before the fully-connected classifier is set to 0.1, and the gradient norm is clipped to be less than 5.0.

### 4.2 TREND OF DIFFERENT CRITERIA

We inspect how BR@K, P@K, and the correlation criteria converge during the training process. We train a parameter sharing model with 17 layers and 16 initial channels on the de-isomorphic NAS-Bench-201 search space. Shown in Fig. 2(a), the speed of convergence is highly different. BR@K converges very fast. P@K converges in around 250 epochs and then even gradually decreases. Meanwhile, linear correlation, Kendall's Tau, and Spearman correlation are still growing till 500 epochs, while the parameter sharing accuracy grows during the whole 1000 epochs. This indicates

Table 1: Supernet training hyper-parameters

| optimizer | SGD | initial LR | 0.05 |
|---|---|---|---|
| momentum | 0.9 | LR schedule | ReduceLROnPlateau[†] |
| weight decay | 0.0005 | LR decay | 0.5 |
| batch size | 512 | LR patience | 30 |
| dropout rate | 0.1 | grad norm clip | 5.0 |

[†]: Learning rate is decayed by 0.5 when the loss stops decreasing for 30 epochs, see `https://pytorch.org/docs/stable/optim.html#torch.optim.lr_scheduler.ReduceLROnPlateau`.

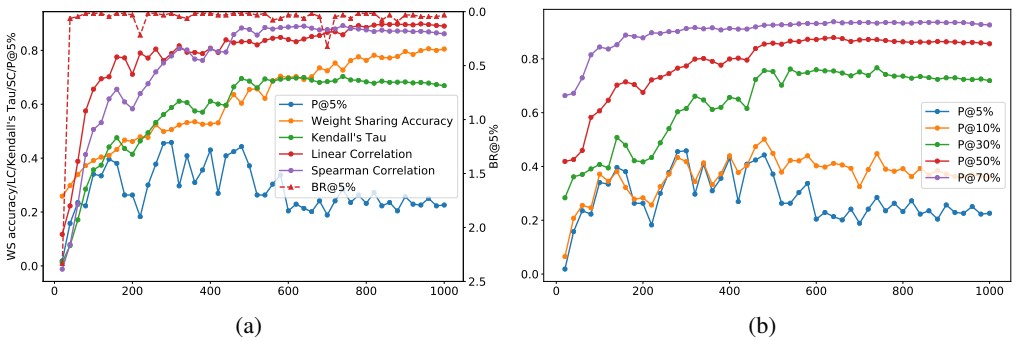

(a)          (b)

Figure 2: Dynamics of different criteria during the supernet training process.

that the models with different rankings change at different speeds as the training progresses, and the top-ranked models stand out faster. Another evidence is shown in Fig. 2(b) that P@5% converges much faster than P@50%. This means that as the training goes on, the supernet is learning to compare architectures with medium or poor performances instead of good ones. Another fact to note in Fig. 2(b) (also see Tab. 3) is that P@5% shows a slow decreasing trend from 200 epochs on. This might be due to that, while the some best architectures stand out very fast in one-shot training, their one-shot performances will be caught up with by others as the training goes on.

### 4.3 SAMPLING STRATEGY

The NAS-Bench-201 search space includes many isomorphic architectures. We expect that one-shot evaluators could handle isomorphic architectures correctly, which means that their evaluated rewards should be close. We calculate the average variances of test accuracy and ranking in isomorphic groups during the training process, as shown in Tab. 2. As the training progresses, the variance

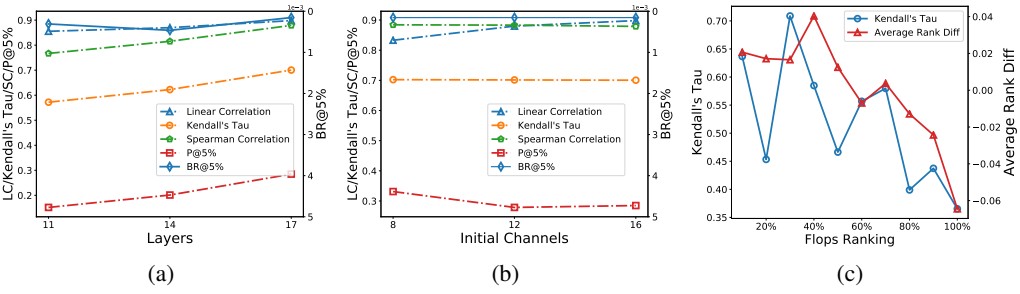

(a)          (b)          (c)

Figure 3: (a) Influence of layer proxy. (b) Influence of channel proxy. (c) Kendall's Tau and average rank difference of architectures in different FLOPs groups.

within the group gradually shrinks, which indicates that more sufficient training makes one-shot evaluator handle isomorphic architectures better.

Table 2: Average standard deviation of accuracies and rankings in architecture isomorphic groups. "GT" stands for the average of the Ground-Truth performance deviations in all isomorphic groups

| epochs | GT | 200 | 400 | 600 | 800 | 1000 |
|---|---|---|---|---|---|---|
| Accuracy std (%) | 0.0392 | 0.193 | 0.161 | 0.0901 | 0.0977 | 0.0911 |
| Global Accuracy std (%) | 12.9 | 11.4 | 16.5 | 13.3 | 21.1 | 21.1 |
| Ranking std | 65.1 | 162 | 153 | 123 | 116 | 117 |

We compare the results of sampling with/without isomorphic architectures during training in Tab. 3. The results show that if de-isomorphism sampling is not used, the supernet performs much worse on good architectures in that BR@1%, BR@5% and P@5% are significantly worse (2.515% V.S. 0.015%, 2.221% V.S. 0.015%, 9.22% V.S. 28.48%). In this case, we find that the top-ranked cell architectures are simple architectures (e.g., a single convolution). That is to say, parameter sharing training without de-isomorphism training might over-estimate simple architectures. This might be due to that the equivalent sampling probability is larger for these simple architectures with many isomorphic counterparts. We also experiment with the post de-isomorphism technique, in which the performances of architectures in each isomorphic group are averaged during testing, while no changes are incorporated in the training process. We find that post de-isomorphism can achieve improvements over "No De-isomorphism". This might owe to the decreased variance of the estimates.

Table 3: Comparison of (no) de-isomorphism sampling in supernet training. "BR@1" indicates the relative ground-truth ranking of the architecture with the highest one-shot score

| Epochs | criterion | 200 | 400 | 600 | 800 | 1000 |
|---|---|---|---|---|---|---|
| No De-isomorphism | BR@1 | 3.372% | 4.493% | 21.555% | 13.139% | 11.379% |
| | BR@1% | 0.012% | 0.019% | 2.854% | 2.739% | 2.515% |
| | BR@5% | 0.006% | 0.013% | 2.336% | 2.310% | 2.221% |
| | P@5% | 39.69% | 12.03% | 6.53% | 7.68% | 9.22% |
| | P@50% | 79.27% | 81.59% | 88.04% | 89.20% | 89.72% |
| | $\tau$ | 0.6033 | 0.6500 | 0.6951 | 0.7032 | 0.7127 |
| De-isomorphism | BR@1 | 1.840% | 2.737% | 5.212% | 5.707% | 2.273% |
| | BR@1% | 0.046% | 0.031% | 0.062% | 0.139% | 0.015% |
| | BR@5% | 0.015% | 0.015% | 0.045% | 0.090% | 0.015% |
| | P@5% | 41.48% | 39.31% | 30.65% | 28.48% | 28.48% |
| | P@50% | 73.03% | 83.76% | 87.44% | 86.39% | 86.45% |
| | $\tau$ | 0.4967 | 0.6735 | 0.7087 | 0.6989 | 0.7005 |
| Post De-isomorphism | BR@1 | 3.866% | 4.964% | 20.260% | 12.465% | 10.950% |
| | BR@1% | 0.015% | 0.015% | 0.479% | 0.634% | 0.247% |
| | BR@5% | 0.015% | 0.015% | 0.340% | 0.217% | 0.093% |
| | P@5% | 46.13% | 37.77% | 23.52% | 21.36% | 24.77% |
| | P@50% | 76.89% | 83.14% | 87.44% | 87.44% | 87.81% |
| | $\tau$ | 0.5511 | 0.6558 | 0.7123 | 0.7109 | 0.7226 |

Tab. 4 shows the comparison of using different architecture sample numbers in supernet training. We can see that using multiple architecture MC samples is not beneficial, and MC sample=1 is a good choice. We also adapt Fair-NAS (Chu et al., 2019) sampling strategy to the NAS-Bench-201 search space (a special case of MC sample 5), and find it does not bring improvements.

## 4.4 PROXY MODEL

Due to memory and time constraints, it is common to use a shallower or thinner proxy model in the search process. The common practice is to search using small proxy models with fewer channels and layers, and then "model augment" the discovered architecture to large neural networks. From the experimental results shown in Fig. 3(a)(b), we can see that channel proxy has little influence while

Table 4: Comparison of using different numbers of architecture Monte-Carlo samples in every super-net training step. Upper: The training epochs of models with 1/3/5 MC samples and Fair-NAS are 1000/333/200/200. Lower: The training epochs of models with 1/3/5 MC samples and Fair-NAS are all 1000. All these results are tested with post de-isomorphism

| Equivalent 1000 epochs | 1 | 3 | 5 | Fair-NAS (Chu et al., 2019) |
|---|---|---|---|---|
| BR@5% | 0.093% | 0.139% | 0.495% | 0.139% |
| P@5% | 24.77% | 9.60% | 20.74% | 11.76% |
| $\tau$ | 0.7226 | 0.7128 | 0.6714 | 0.7137 |
| 1000 epochs | 1 | 3 | 5 | Fair-NAS (Chu et al., 2019) |
| BR@5% | 0.093% | 0.015% | 0.124% | 0.031% |
| P@5% | 24.77% | 14.24% | 17.03% | 15.17% |
| r $\tau$ | 0.7226 | 0.7025 | 0.7018 | 0.6965 |

layer proxy reduces the reliability of search results. Thus, for cell-based search spaces, proxy-less search w.r.t the layer number is worth studying.

### 4.5 OVER- AND UNDER-ESTIMATION OF ARCHITECTURES

For one-shot evaluators, we expect that the training process is fair and balance for all architectures. However, sub-architectures have different amounts of calculation, and they might converge with a different speed. To understand which architectures are under- or over-estimated by the one-shot evaluators, we inspect the Ranking Diff of the ground truth performance and the one-shot evaluation of an architecture $a_i$: $r_i - n_i$. We divide the architectures into ten groups according to the amount of calculation (FLOPs), and show Kendall's Tau and Average Rank Diff of each group in Fig. 3(c).

Note that a positive Ranking Diff indicates that this architecture is over-estimated, otherwise it is underestimated. The x-axis is organized such that the architecture group with the least FLOPs is at the leftmost. The average Rank Diff shows a decreasing trend, which means that the larger the model, the easier it is to be underestimated. Also, the decreasing intra-group Kendall's Tau indicates that it is harder for the one-shot evaluator to compare larger models (which usually have better performances) than comparing smaller models.

## 5 PREDICTOR-BASED EVALUATORS

In this section, we employ the same criteria (i.e., Kendall's Tau, Precision@K, BestRanking@K) to assess the architecture predictors (i.e., predictor-based evaluators).

### 5.1 EXPERIMENTAL SETUP

We experiment with four different architecture predictors: MLP, LSTM, GATES (Ning et al., 2020), and a random forest regressor (RF). For MLP, LSTM, and RF, we serialize each architecture matrix using the six elements of its lower triangular portion. We follow (Ning et al., 2020) to construct MLP, LSTM, and GATES. The RF predictor applies a random forest regressor on the 6-dim sequence.

For optimizing MLP, LSTM, and GATES, an ADAM optimizer with a learning rate of 1e-3 is used, the batch size is 512, and the training lasts for 200 epochs. Following (Ning et al., 2020), a hinge pairwise ranking loss with margin 0.1 is used for training these predictors. For RF, we use a random forest with 100 CARTs to predict architecture performances.

### 5.2 EVALUATION RESULTS

We train these predictors on training sets of different sizes: 39 (0.25%), 78 (0.5%), 390 (2.5%), 781 (5%). Specifically, for each training set size, we randomly sample three different training sets, and train each predictor on each training set with three different random seeds (20, 2020, 202020). After training, we evaluate each model on the whole NAS-Bench-201 search space using Kendall's Tau,

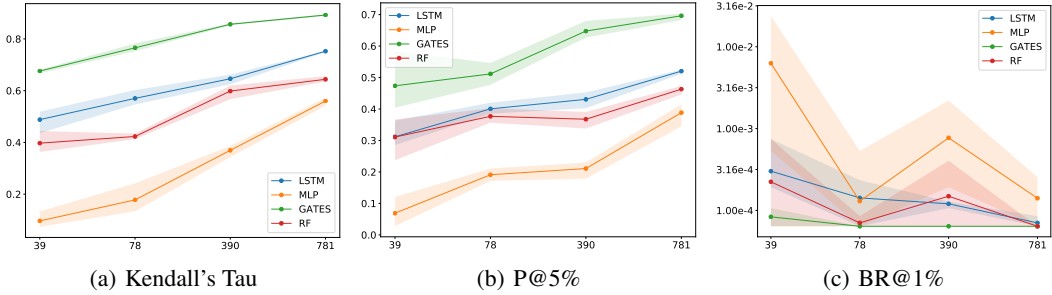

(a) Kendall's Tau  (b) P@5%  (c) BR@1%

Figure 4: Comparison of predictor performances. x axis: Different training set size. Each line contains the results of 3 differently randomly sampled training set. The results of using different training seeds are averaged.

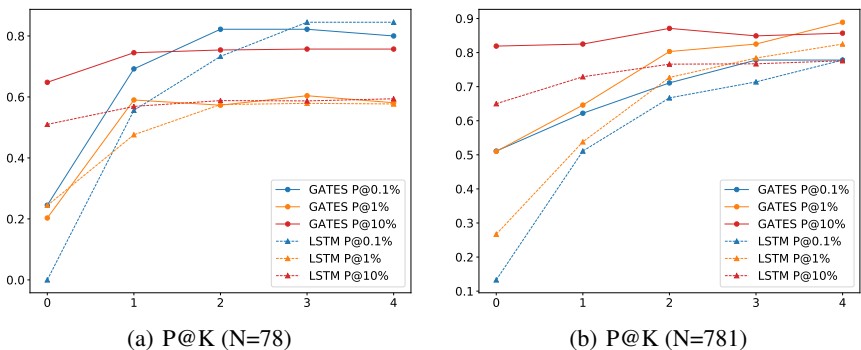

(a) P@K (N=78)  (b) P@K (N=781)

Figure 5: Comparison of predictors after training of 5 stages. In each stage, $N$ architecture is chosen, evaluated, and used to train the predictor along with previous architecture data.

Precision@K, and BestRanking@K. Different from (Ning et al., 2020), the evaluation is carried out on all the architectures, instead of a separate validation split. As shown in Fig. 4, GATES outperforms other predictors in all settings.

As can be seen, training with different seeds on different training sets leads to similar results. In contrast, we found that training predictors with regression loss is not stable and sensitive to the choice of the training set. For example, the Kendall's Taus of 3 GATES models trained using the regression loss on different training sets of size 78 are 0.7135, 0.7240, and 0.2067, respectively, while with ranking loss, the results are 0.7597, 0.7750, and 0.7645, respectively. These additional results are listed in the Appendix.

As illustrated in Fig. A1 in the Appendix, there are usually multiple stages in a predictor-based NAS search process. In each stage $i$, the current predictor is used to choose $N$ architectures to be evaluated by the oracle evaluator, then the results will be used to update the predictor. In our experiment, we select the top-$N$ architectures predicted by the current predictor in the whole search space. As shown in Fig. 5, P@0.1% increases a lot after multiple stages of training, which indicates that the predictors perform better and better on distinguishing the good architectures. This is expected since the training data are more concentrated on the good architectures. Also, the performance gap between different predictors (LSTM/GATES) shrinks as more training data become available. The Kendall's Tau, BR@K, and performance distribution results of multiple stages are listed in the Appendix.

## 6  CONCLUSION

This paper assesses parameter-sharing evaluators and architecture predictors on the NAS-Bench-201 search space, with a set of carefully developed criteria. Hopefully, knowledge revealed by this paper can guide future NAS application and motivate further research.

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

# A  ADDITIONAL RESULTS ABOUT PREDICTOR-BASED EVALUATORS

## A.1  PREDICTOR PERFORMANCES RESULTS: SINGLE- AND MULTIPLE-STAGE

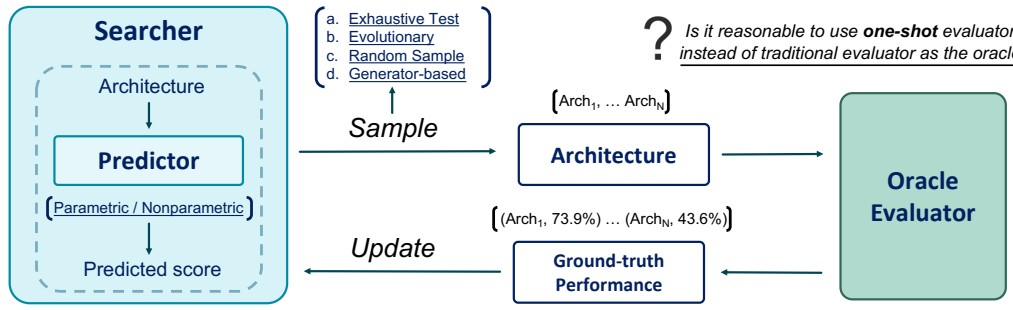

Figure A1: The overview of predictor-based neural architecture search (NAS). The underlined descriptions between the parenthesis denote different methods.

Table A1: The Kendall's Tau of different predictors on 3 different randomly sampled training dataset of size 78

| Training Loss | Ranking | | | Regression | | |
|---|---|---|---|---|---|---|
| Dataset | 1 | 2 | 3 | 1 | 2 | 3 |
| MLP | 0.1330±0.074 | 0.1560±0.0078 | 0.2481±0.0069 | 0.0111±0.0000 | 0.0548±0.0276 | 0.0467±0.0130 |
| LSTM | 0.5631±0.0060 | 0.6028±0.0457 | 0.5487±0.0150 | 0.6024±0.0039 | 0.5784±0.0180 | 0.4656±0.0176 |
| GATES (Ning et al., 2020) | 0.7597±0.0079 | 0.7750±0.0106 | 0.7645±0.0054 | 0.2067±0.0000 | 0.7240±0.0074 | 0.7135±0.0055 |
| RF (Sun et al., 2019) | - | - | - | 0.4329±0.0077 | 0.4123±0.0104 | 0.4218±0.0119 |

Table A2: The performance distribution, BR@K, Kendall's Tau of 5 training stages. In each stage, $N = 78$ architectures are chosen, evaluated, and used to train the predictor along with previous architecture data. Note that in this table, K in BR@K is the absolute architecture number without normalization

| | Stage | 0 | 1 | 2 | 3 | 4 |
|---|---|---|---|---|---|---|
| GATES | Perf. Range | [0.560, 0.938] | [0.921, 0.944] | [0.935, 0.944] | [0.933, 0.944] | [0.933, 0.944] |
| | Perf. Std | 6.43e-2 | 4.59e-3 | 2.16e-3 | 2.18e-3 | 2.30e-3 |
| | BR@11/BR@7/BR@1 | 1/2/306 | 1/1/3 | 1/1/2 | 1/1/3 | 1/1/3 |
| | Kendall's Tau | 0.769 | 0.759 | 0.752 | 0.742 | 0.725 |

| | Stage | 0 | 1 | 2 | 3 | 4 |
|---|---|---|---|---|---|---|
| LSTM | Perf. Range | [0.560, 0.938] | [0.922, 0.944] | [0.922, 0.944] | [0.932, 0.944] | [0.934, 0.944] |
| | Perf. Std | 6.43e-2 | 4.52e-3 | 2.63e-3 | 2.42e-3 | 1.98e-3 |
| | BR@11/BR@7/BR@1 | 99/268/393 | 2/2/9 | 1/1/6 | 1/2/5 | 1/1/3 |
| | Kendall's Tau | 0.562 | 0.556 | 0.571 | 0.739 | 0.724 |

## A.2  OVER- AND UNDER-ESTIMATION OF ARCHITECTURES

Fig. A2(d)(e)(f) illustrates the relationship between the FLOPs of architectures and how it is likely to be over-estimated. It seems that MLP and RF are more likely to overestimate the smaller architectures and underestimate the larger ones, while LSTM and GATES show no obvious preference on the architectures' FLOPs. Fig. A2(a)(b)(c) shows that GATES can give more accurate rankings on smaller architectures than larger architectures, which indicates that GATES might still have trouble in comparing larger architectures that usually have good performances.

## A.3  ONE-SHOT ORACLE EVALUATOR

Luo et al. (2018) made an attempt to use a parameter-sharing evaluator as the oracle evaluator in Fig. A1. That is to say, they use the noisy signals provided by the parameter sharing evaluator to train

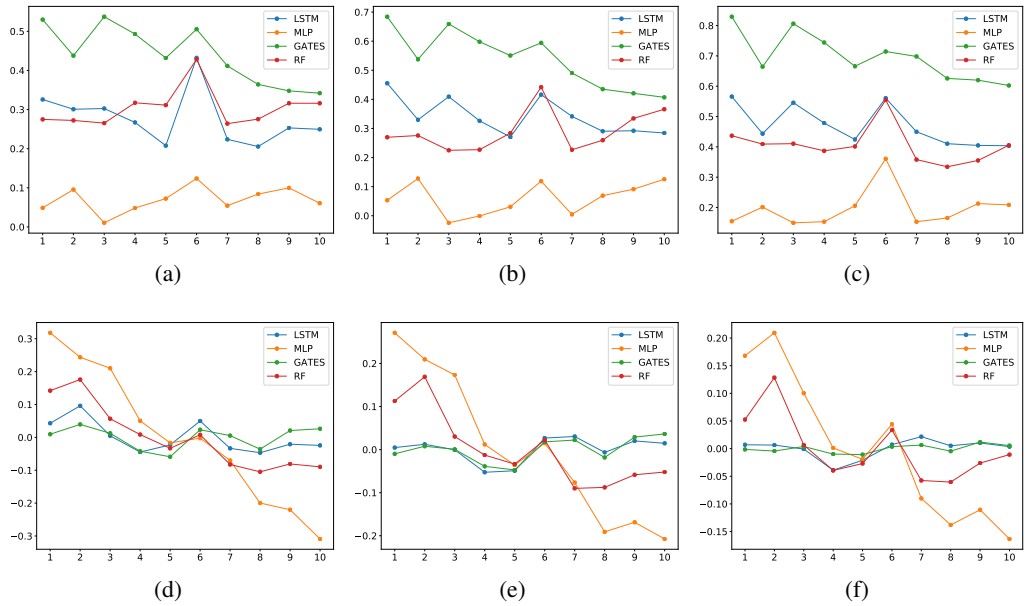

Figure A2: (a)(b)(c) Kendall-tau in different FLOPs groups, the training set size is 39, 78 and 390, respectively. (d)(e)(f) Average rank difference in different FLOPs groups, the training set size is 39, 78 and 390, respectively.

the predictor. This will significantly accelerate the NAS process, compared with using an expensive traditional evaluator. However, it is found to cause the NAS algorithm to fail to discover good architectures. Also, predictors have been used to accelerate parameter-sharing NAS methods (Li et al., 2020; Wang et al., 2020), since one predictor forward pass is faster than testing on the whole validation queue, even if no separate training phase is needed. In this section, we explore whether a predictor can recover from the noisy training signals provided by the parameter-sharing evaluator. Since GATES achieves consistently better results than other predictors, it is used in the following experiments. Specifically, we want to answer two questions:

1. Can sampling only a subset during supernet training help achieve better one-shot Kendall's Tau on these architectures?

2. Can predictor training help recover from the noisy training signals provided by the one-shot evaluator?

We randomly sample 78 architectures from the search space. Two differently trained parameter-sharing evaluators are used to provide the one-shot instruction signal of these 78 architectures: 1) Uniformly sampling from the whole search space, 2) Uniformly sampling from the 78 architectures. We find that strategy 1 (sampling from the whole search space) can get a higher evaluation Kendall's Tau, no matter whether the evaluation is on the 78 architectures (0.657 V.S. 0.628) or the whole search space (0.701 V.S. 0.670). Thus the answer to Question 1 is "No".

Then, to answer the second question, we utilize the one-shot instruction signal provided by the supernet trained with 15625 architectures to train the predictor[1]. The Kendall's Tau between the architecture scores given by the resulting predictor and the ground-truth performances is 0.718 on all the 15625 architectures, which is slightly worse than the one-shot instruction signals (0.719). More importantly, BR@1% degrades from 2.5% to 12.1%, thus the answer to Question 2 is "No'.

Thus, we conclude that although training a predictor using one-shot signals can bring acceleration, since no extra inference is needed during the search, it is not beneficial in the sense of evaluation quality (especially of good architectures). Perhaps, incorporating more manual prior knowledge and regularizations can increase the denoising effect, which might be worth future research.

---

[1]The average of scores provided by 3 supernets trained with different seeds is used.

## B    ADDITIONAL DISCUSSION AND RESULTS ABOUT ONE-SHOT EVALUATORS

### B.1    DISCUSSION ON DE-ISOMORPHIC SAMPLING

To conduct isomorphic sampling, we first design a simple encoding method, which can canonicalize computationally-isomorphic architectures to the same string and non-isomorphic architectures to different strings. In our paper, we use the encoding method to find out all isomorphic groups in the search space and make them as a table. Then, during the supernet training process, we sample architecture groups from this table uniformly. This method is feasible since the benchmark search space is not large. In practice, one can use lazy table-making and rejection sampling to conduct de-isomorphic sampling, by only accepting new or representative architecture samples. More specifically, one first encodes each sampled architecture into a canonical string. If this canonical string has not appeared before, this string stands for a new isomorphic group, and the architecture is recorded as the representative architecture for this isomorphic group. This architecture sample is also accepted. If this canonical string has been recorded before, we only accept the architecture sample if it is the representative architecture for its canonical string.

The simple encoding method is similar to the one in NASBench-201. To be more specific, the encoding method goes as follows. Denoting the expression of the $i$-th node as $S_i$, and the operation in the directed edge (j,i) as $A_{ji}$, the expression $S_i$ can be written as

$$S_i = \text{Concat}(\text{Sort}(\{ \text{``(''} + S_j + \text{``)''} + \text{``\%''} + A_{ji}\}_{j \in P(i)})),$$

where $P(i)$ denotes the set of predecessor nodes of $i$, and Sort sorts the strings in dictionary order. We calculate $S_i(i = 1, \cdots, 4)$ in topological order, and the expression $S_4$ at the final output node is used as the encoding string of the architecture.

### B.2    RESULTS ON DE-ISOMORPHIC SAMPLING

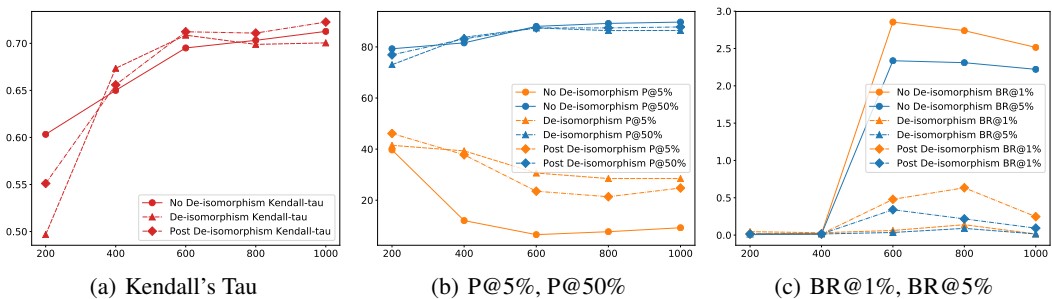

(a) Kendall's Tau          (b) P@5%, P@50%          (c) BR@1%, BR@5%

Figure A3: How the evaluation quality evolves along the supernet training process with different sampling strategy. The data is also listed in Tab. 3 in the main text.

Fig. A3(a) shows that the ranking correlation keeps increasing throughout the training process. Fig. A3(b) indicates that P@50% increases with more sufficient training, while P@5% slightly shrinks, especially the supernet trained without de-isomorphism sampling. As also been analyzed in the main text, this might arise from the fact that simple architectures have more isomorphic counterparts. Fig. A3(c) shows that without de-isomorphic sampling, BR@1% and BR@5% deteriorate in the late stages of the training process.

