# OpenReview forum: "A Surgery of the Neural Architecture Evaluators"
_ICLR.cc/2021/Conference — Reject_

### Official Review · AnonReviewer4 · 2020-10-14
**Interesting benchmark paper, some concerns about experiment methodology**

**Rating:** 3
**Confidence:** 3

**Review:**

## Post-Rebuttal Update
Since many of the concerns from my original review would require a major revision to address, I've tentatively left my original review score (3) unchanged. The biggest reason is that I'm still concerned that insufficient hyper-parameter tuning could lead to wrong conclusions; this would need more work to address. I've included some additional notes below.

In their response to my review and those of the other reviewers, the authors promise to address a number of issues with the current draft. While I hope these updates will ultimately improve the paper, the authors' current responses don't provide me with enough information merit increasing my ICLR review score. For example:
* "We will try to auto-tune the hyper-parameters of different predictors in the future."
* "We think we can improve Figure 3c" by measuring mean/variance across multiple runs.

On a positive note: I'd like to thank the authors for clearing up my confusion about ranking diff under "additional notes". I also think the authors' explanation of their isomorphic sampling procedure in the response to AnonReviewer2 addresses one of my questions.

## Recommendation
I like the submission's high-level goal of trying to do careful, controlled studies of different factors which can affect a NAS algorithm's ability to ability to rank different candidate architectures within a search space. And the paper seems reasonably well-organized and well-written. However, due to some concerns about the paper's methodology and conclusions (discussed below), I do not feel comfortable recommending the paper for acceptance in its current form.

## Background
The goal of Neural Architecture Search (NAS) is to automatically identify network architectures from a human-defined search space which have good accuracy/speed tradeoffs. However, ranking the accuracies of different candidate architectures within a search space can be extremely compute-intensive if done naively, by training each candidate network from scratch and then evaluating it.

The submission conducts empirical studies on the NASBench-201 benchmark dataset for two previously proposed classes of techniques which can be used to efficiently rank different network architectures within a search space without training all of them from scratch:
1. train a one-shot model -- a single shared set of weights which can be used to evaluate many different candidate architectures.
2. train a predictor by (a) sampling a limited number of candidate architectures from the search space, (b) training/evaluating each one, and then (c) using the resulting <architecture, accuracy> pairs to train a regression model which can predict the accuracy of *any* architecture within the search space.

## Paper Contributions
The submission conducts experiments to estimate how changes in training/evaluation setups will affect the ability of these two methods to rank different candidate architectures within a search space.

For one-shot models:
* Evaluating how the one-shot model's ability to rank the most accurate architectures in the search space vs. ranking random architectures evolves over the course of a search.
* Evaluating two tricks for reducing the cost of training a one-shot model: (a) reducing the number of layers in the one-shot model, or (b) reducing the number of filters per layer.
* Changing the architecture sampling strategy that's used when training the one-shot model.
* "Post de-isomorphism:" The NASBench-201 search space has multiple ways to encode the same architecture which produce different accuracies according to a one-shot model; here, the authors propose taking the average accuracy over all possible encodings of a given architecture.
* Some experiments to quantify the predictor's ability to rank large vs. small models.

For predictors/regressors:
* Comparing the relative ranking ability of predictor models trained using random forests regressors, MLPs, LSTMs, and GATES.

## Concerns about methodology
Many of the paper's experiments -- especially in earlier sections -- seemed reasonable. In particular: "the influence of channel vs layer proxy" (Figure 3) and the effects of post-de-isomorphism (Table 3) and sampling (Table 4) on one-shot model / supernet rankings. However, I have concerns that about a number of experiments, especially in later sections of the paper:

**Insufficiently tuned hyper-parameters?**
In Section 5.1, when describing their experiments comparing different predictors/regressors, the authors write that "For optimizing MLP, LSTM, and GATES, an ADAM optimizer with a learning rate of 1e-3 is used, the batch size is 512, and the training lasts for 200 epochs." This makes me concerned that (i) the authors used a default set of hyper-parameters to optimize their models, and (ii) few attempts were made to tune the hyper-parameters. Because the quality of models trained using gradient descent can be quite sensitive to the training hyper-parameters, this makes me wonder whether the authors would've drawn different conclusions about the performance of these predictors if they'd spent more time on hyper-parameter tuning. For example: experimenting with learning rate schedules, tuning the learning rate and weight decay used during training, tuning the MLP's hidden layer sizes, etc.

The authors also claim that "training predictors with regression loss is not stable and sensitive to the choice of training set." If the authors' hyper-parameters were not sufficiently well-tuned, that could explain the instabilities the authors saw. Other standard tricks (e.g., normalizing the labels to have mean 0 and variance 1 before performing a regression) could also drastically affect this conclusion.

**Seemingly trivial results presented as non-trivial?**
In Section 4.5, the submission argues that ""the average rank diff shows a decreasing trend, which means that the larger the model, the easier it is to be underestimated." And the intro states that "parameter sharing evaluator tends to over-estimate smaller architectures." These findings are presented in a way which suggests that there's bias in the one-shot model itself, but the observation itself seems trivial, and would likely hold even if the one-shot model was unbiased. To take an extreme example: the one-shot model can underestimate but cannot overestimate the rank of the most accurate architecture in the search space. So the expected rank of this top architecture would almost certainly be less than the true rank unless the one-shot model provided a perfect ranking of architectures within the search space.

**Unsupported claims related to multi-stage training?**
In Section 5.2: the authors claim that "As shown in Fig. 5, P@0.1% increases a lot after multiple stages of training, which indicates that the predictors perform better and better on distinguishing good architectures. This is expected since the training data are more and more concentrated on good architectures."

If I understand this figure correctly, the authors are using progressively more training data at each stage. In this case, the improvements could simply be explained by the fact that they're using a larger training dataset, and have nothing to do with the fact that the training data is concentrated on good architectures. Additional baselines would be needed to test the authors' original hypothesis.

## Additional notes

**Missing description of de-isomorphic sampling?**
I had trouble finding details about how de-isomorphic sampling was performed during one-shot model training. (Please let me know if there's a description I overlooked.)

**End of Section 4.2:** "with medium or pool performance:" I think this is a typo. Did you mean "poor" rather than "pool"

**Section 4.5:** "We inspect the Ranking Diff of the ground truth performance and the one-shot evaluation of an architecture [...] r_i - n_i. [...] Note that a positive Ranking Diff indicates that this architecture is over-estimated, otherwise it is underestimated." I interpreted "diff of the ground truth performance and the one-shot performance" to mean "[ground truth performance] minus [one-shot performance]". But under this interpretation, I think a *negative* ranking diff would mean that the architecture is over-estimated by the one-shot model. Did you mean to write "ground truth diff of the one-shot evaluation and the ground truth evaluation" rather than the other way around?

---

### Official Review · AnonReviewer3 · 2020-10-23
**Review of "A Surgery of the Neural Architecture Evaluators"**

**Rating:** 5
**Confidence:** 4

**Review:**

### Contributions ###
* The paper conducts an extensive study of efficient neural architecture "evaluators", in particular of one-shot evaluators with shared weights and of predictor-based evaluators.
* Several findings are extracted from the empirical evaluation on the NasBench-201 benchmark. To summarize some key findings: (a) proxy models need not be wide but should be deep (channel proxy works better than layer proxy), (b) one-shot evaluators with parameter-sharing tend to overestimate smaller architectures and are also better at comparing these, (c) predictors benefit from being trained with a ranking loss rather than regression loss.

### Significance ###
Scaling neural architecture search (NAS) to large-scale tasks requires efficient ways of estimating architecture performance. Several different, more-or-less efficient ways of estimating architecture performance have been proposed; it thus becomes increasingly important to understand errors and biases of these different estimators. This work provides very valuable insights regarding these points for the most promising classes of architecture performance estimators. It thus can guide and motivate future directions for efficient performance estimation for NAS, and is thus a significant contribution.
Significance would be strengthened by releasing code for the conducted experiments. This would allow future methods be  easily compared to the results obtained in this paper.

### Originality ###
There are some prior works also studying quality of efficient architecture performance estimators (summarized by the authors in Section 2.1). However, the proposed work goes considerably beyond the scope of these works. It heavily builds upon NasBench-201. However, I consider the paper still sufficiently original for a publication at ICLR.

### Clarity ###
Generally, the summary of the conducted studies and the presentation of the results is clear and easy to follow, with two exceptions: (1) Figure 2a shows too many different things in one plot and is hard to grasp; I think splitting this plot into several independent ones would be helpful. (2) Table 3 would best be represented graphically rather than as a table.
In terms of wording, it would make sense in my opinion to use some established terms and concepts from statistics. For instance, the models studied in this paper dot not really "evaluate" performance but estimate it; thus calling them "estimator" would be more appropriate than "evaluator". Estimators might have a bias ("performance of small models is typically overestimated"), and their variance might vary for different parts of the search space ("ranking smaller models is easier than ranking larger ones).

I would ask the authors to clarify which architecture is used for GT column in Table 2. A randomly sampled one?
Moreover, does dropout in Table 1 actually refer to dropout or rather to the more common Droppath?

### Quality ###
There are some evaluations for which I am not totally convinced that the drawn conclusions are trustworthy:
* Stating that the ranking loss is more stable than regression loss at the end of Section 5.2 seems too far stretched given that there is only 1 outlier in 3 runs where regression loss failed. For drawing this conclusion, far more repetitions and a statistical significance test would be required.
* In general, having error bars on the reported results (from several repetitions of the same setting) and checking for statistical significance would increase quality of the evaluation considerably. For example, effects in Figure 3c are rather small and noisy and the trend might in the worst case be due to chance.
* Table 2 indicates a decrease of variance for estimated accuracy of isomorphic architectures for longer training. However, couldn't this be an indication that all architectures converge to high accuracy and thus total variance shrinks (and not only those of isomorphic ones?) I would ask the authors to also report global accuracy standard deviation (or normalize by it)
* Generally, the quality of a ranking depends not only on the strength of an estimator but also on the scale of differences in performances. In the extreme case, two architectures have the same performance; how should these be ranked? A discussion of this and how it is addressed would be helpful.
* The fact that post de-isomorphism works surprisingly well is stated but not explained. An attempt at explaining this observation would be highly appreciated. For instance: post de-isomorphism averages performance estimates of isomorphic architectures. By this, it should also decrease variance of the estimates. Could this explain the improved ranking quality?
* How specific are the obtained results for the hyperparameters used (Table 1) and the benchmark? For drawing general conclusions, it would be required testing also other hyperparameters and on other datasets/benchmarks.

### Recommendation ###
To summarize, I think this paper has the clear potential for getting accepted at ICLR. However, in the current form, there are several issues that would need to be fixed. For the time being, I lean towards rejection but I would be willing to increase my score if these points would be addressed.

### Final Recommendation after Author Response ###
I have read the author response and appreciate their feedback. As the authors could not address some of the issues (error bars/statistical significance testing, other hyperparameters and other datasets/benchmarks) in the restricted time of the rebuttal period, I will keep my rating and recommend rejecting this submission for ICLR. However, I also encourage the authors to resubmit a revised version of the paper taking all feedback into account since I see clear potential.

---

### Official Review · AnonReviewer2 · 2020-10-27

**Rating:** 4
**Confidence:** 4

**Review:**

# Summary

The paper assesses two different approaches to speed up the evaluations of neural network architectures for neural architecture search (NAS). The first one is weight sharing, which trains a supernetwork that contains all possible architecture of the search space. The performance of single architectures can be then approximated by simply using the shared parameters of the supernetwork. The second approach is to use different kind of predictors that are trained on offline evaluated architectures. Several methods following these two approaches from the literature are evaluated on the NASBench201 benchmark based on different rank-based evaluation scores.


In general I think the paper addresses an interesting problem in neural architecture search. However, compared to existing work, the new insights that the paper presents are rather limited. Particularly, since the results are only based on a single benchmark. I therefore don't think that the paper reaches the bar of acceptance.


# pros

- The main contribution of the paper to provide a detailed analysis and comparison of different evaluation strategies for neural architecture search.

- The paper is well written and easy to follow. Also the usage of the different metrics is well motivated and appears reasonable.


# cons


 - The paper only evaluates on the NASBench201 benchmark, which consists of the same set of architectures evaluated on three different datasets. However, it is unclear how reliable conclusions are and if they also transfer to other search spaces. At least other tabular benchmarks, e.g NASBench101 and NASBench101-shot could be added to the evaluation. Even better would be to also include a larger non-tabular search space (e.g DARTS), which, due to the computational requirements, could be then less exhaustive.


- The novelty of the paper seems to be rather small.  Ning et. al. also compared different predictor (MLP, LSTM, GATES) on  Nasbench201 benchmarks. They even report the same metrics (e.g Precision@K). It seems that the only new contribution of this paper is to extend this comparison by including a random forest predictor, which is arguably a rather simple baseline.

-Also, the correlation between shared weights of the one-shot model and the architectures from the search space has been investigated several times before (for example Dong et. al., Yang et. al.). Unfortunately, the paper  does not state  which new insights it provides  that extends the contributions of previous papers.


# Questions for rebuttal

- How does the de-isomorphic sampling works in general?

- Could be that the ranking of top performing architectures is low because the difference between architecture are almost the same?

- How exactly is the parameter sharing model in Section 4.2 trained on the de-isomorphic NAS-Bench201 search space? How does it differ from training on the full search space?



# minor comments / typos

- Figure 3: I think caption and plot a) and b) are mixed up.

- Section 4.2 pool -> poor


# Post Rebuttal

I thank the authors for replying to my question. Unfortunately, the authors did not present any new results on other benchmarks (e.g Nasbench101-shot).  I will therefore keep my score.

---

### Official Review · AnonReviewer1 · 2020-11-09
**Evaluations of Paper "A Surgery of the Neural Architecture Evaluators"**

**Rating:** 5
**Confidence:** 3

**Review:**

The authors assess both the parameter-sharing evaluators and predictor-based evaluators on the NAS-Bench-201 search space, with a set of NAS-oriented criteria to understand the behavior of fast architecture evaluators in different training stages. Experimental results on the CIFAR-10 dataset using the NAS-Bench-201 search space are reported.

Strengths:

The authors conduct an extensive assessment of fast architecture evaluators on the NAS-Bench-201 search space, and provide insights on how different configurations and strategies could influence the fitness of the evaluators.

Weaknesses:

(1) This paper is focused on the assessment of existing fast architecture evaluators, rather than the proposal of new methods or architecture evaluators. It aims to share experiences on neural architecture evaluations. The overall novelty of this work may not be very strong.

(2) The experiments are conducted on a single dataset CIFAR-10. It is not clear if the provided suggestions can be generalized to a broad set of machine learning tasks.

Additional comments:

The parameter-sharing evaluators and predictor-based evaluators are evaluated separately under different experiment setups. I think it would be interesting to compare the performance of all the evaluators on the same task. This may provide further insights as well as concrete examples on the pros and cons of these two types of evaluators.

---

### Decision · Program_Chairs · 2021-01-07
**Final Decision**

**Decision:**

Reject

**Comment:**

This paper presents an empirical study of different efficient ways to estimate the performance of architectures in NAS, focussing on weight sharing and performance prediction methods.
Most reviewers appreciated the paper's goal of performing a careful, controlled study of different factors that can affect the ranking of architectures. However, all reviewers also had substantial concerns. Many of these could in principle be fixed by additional experiments, but the short time window of the author response period did not allow for this.
As a result, all reviewers voted for rejection, and I will follow that recommendation.
Nevertheless, I would like to encourage the authors to continue this work, as I believe that the NAS community needs more careful controlled studies of this type. For their next version, I encourage the authors to take into account the many points mentioned by the reviews.